# Hierarchical VAE with a Diffusion-based VampPrior

**Anna Kuzina**                                                     *a.kuzina@vu.nl*
*Department of Computer Science,*
*Vrije Universiteit Amsterdam, Netherlands.*

**Jakub M. Tomczak**                                              *j.m.tomczak@tue.nl*
*Department of Mathematics and Computer Science,*
*Eindhoven University of Technology, Netherlands.*

**Reviewed on OpenReview:** *https://openreview.net/forum?id=NUkEoZ7Toa*

## Abstract

Deep hierarchical variational autoencoders (VAEs) are powerful latent variable generative models. In this paper, we introduce Hierarchical VAE with Diffusion-based Variational Mixture of the Posterior Prior (VampPrior). We apply amortization to scale the VampPrior to models with many stochastic layers. The proposed approach allows us to achieve better performance compared to the original VampPrior work and other deep hierarchical VAEs, while using fewer parameters. We empirically validate our method on standard benchmark datasets (MNIST, OMNIGLOT, CIFAR10) and demonstrate improved training stability and latent space utilization.

## 1 Introduction

Latent variable models (LVMs) parameterized with neural networks constitute a large group in deep generative modeling (Tomczak, 2022). One class of LVMs, Variational Autoencoders (VAEs) (Kingma & Welling, 2014; Rezende et al., 2014), utilize amortized variational inference to efficiently learn distributions over various data modalities, e.g., images (Kingma & Welling, 2014), audio (Van Den Oord et al., 2017), or molecules (Gómez-Bombarelli et al., 2018). The expressive power of VAEs can be improved by introducing a hierarchy of latent variables. The resulting hierarchical VAEs, such as ResNET VAEs (Kingma et al., 2016), BIVA (Maaløe et al., 2019), very deep VAE (VDVAE) (Child, 2021), or NVAE (Vahdat & Kautz, 2020) achieve state-of-the-art performance on images in terms of the negative log-likelihood (NLL). However, hierarchical VAEs are known to have training instabilities (Vahdat & Kautz, 2020). To mitigate these issues, various *tricks* were proposed, such as gradient skipping (Child, 2021), spectral normalization (Vahdat & Kautz, 2020), or softmax parameterization of variances (Hazami et al., 2022). In this work, we propose a different approach that focuses on two aspects of hierarchical VAEs: (i) the structure of latent variables, and (ii) the form of the prior for the given structure. We introduce several changes to the architecture of parameterizations (i.e. neural networks) and the model itself. As a result, we can train a powerful hierarchical VAE with gradient-based methods and ELBO as the objective without any *hacks*.

In the VAE literature, it is a well known fact that the choice of the prior plays an important role in the resulting VAE performance (Chen et al., 2017; Tomczak, 2022). For example, VampPrior (Tomczak & Welling, 2018), a form of the prior approximating the aggregated posterior, has been shown to consistently outperform VAEs with a standard prior and a mixture prior. In this work, we extend the VampPrior to deep hierarchical VAEs in an efficient manner. We propose utilizing a non-trainable linear transformation like Discrete Cosine Transform (DCT) to obtain pseudoinputs. Together with our architecture improvements, we can achieve state-of-the-art performance, high utilization of the latent space, and stable training of deep hierarchical VAEs.

The contributions of the paper are the following:

- We propose a new VampPrior-like approximation of the optimal prior (i.e., the aggregated posterior) which can efficiently scale to deep hierarchical VAEs.

- We propose a latent aggregation component that consistently improves the utilization of the latent space of the VAE.

- Our proposed hierarchical VAE with the new class of priors outperforms other hierarchical VAE models (without additional data augmentations).

## 2 Background

### 2.1 Hierarchical Variational Autoencoders

Let us consider random variables $\mathbf{x} \in \mathcal{X}^D$ (e.g. $\mathcal{X} = \mathbb{R}$). We observe $N$ $\mathbf{x}$'s sampled from the empirical distribution $r(\mathbf{x})$. We assume that each $\mathbf{x}$ has $L$ corresponding latent variables $\mathbf{z}_{1:L} = (\mathbf{z}_1, \ldots, \mathbf{z}_L)$, where $\mathbf{z}_l \in \mathbb{R}^{M_l}$ and $M_l$ is the dimensionality of each variable. We aim to find a latent variable generative model with unknown parameters $\theta$, $p_\theta(\mathbf{x}, \mathbf{z}_{1:L}) = p_\theta(\mathbf{x}|\mathbf{z}_{1:L})p_\theta(\mathbf{z}_{1:L})$.

In general, optimizing latent-variable models with nonlinear stochastic dependencies is non-trivial. A possible solution is approximate inference, e.g., in the form of variational inference (Jordan et al., 1999) with a family of variational posteriors over latent variables $\{q_\phi(\mathbf{z}_{1:L}|\mathbf{x})\}_\phi$. This idea is exploited in Variational Auto-Encoders (VAEs) (Kingma & Welling, 2014; Rezende et al., 2014), in which variational posteriors are referred to as encoders. As a result, we optimize a tractable objective function called the *Evidence Lower BOund* (ELBO) over the parameters of the variational posterior, $\phi$, and the generative part, $\theta$, that is:

$$\mathbb{E}_{r(\mathbf{x})}\left[\ln p_\theta(\mathbf{x})\right] \geq \mathbb{E}_{r(\mathbf{x})}\left[\mathbb{E}_{q_\phi(\mathbf{z}_{1:L}|\mathbf{x})} \ln p_\theta(\mathbf{x}|\mathbf{z}_{1:L}) - D_{\mathrm{KL}}\left[q_\phi(\mathbf{z}_{1:L}|\mathbf{x})\|p_\theta(\mathbf{z}_{1:L})\right]\right], \tag{1}$$

where $r(\mathbf{x})$ is the empirical data distribution. Further, to avoid clutter, we will use $\mathbb{E}_{\mathbf{x}}\left[\cdot\right]$ instead of $\mathbb{E}_{r(\mathbf{x})}\left[\cdot\right]$, meaning that the expectation is taken with respect to the empirical distribution.

### 2.2 VampPrior

The latent variable prior (or marginal) plays a crucial role in the VAE performance, which motivates the usage of data-dependent priors. Note that the KL-divergence term in the ELBO (see Eq. 1) can be expressed as follows (Hoffman & Johnson, 2016):

$$\mathbb{E}_{\mathbf{x}} D_{\mathrm{KL}}\left[q_\phi(\mathbf{z}|\mathbf{x})\|p(\mathbf{z})\right] = D_{\mathrm{KL}}\left[q_\phi(\mathbf{z})\|p(\mathbf{z})\right] + \mathbb{E}_{\mathbf{x}}\left[D_{\mathrm{KL}}\left[q_\phi(\mathbf{z}|\mathbf{x})\|q(\mathbf{z})\right]\right]. \tag{2}$$

Therefore, the optimal prior that maximizes the ELBO has the following form:

$$p^*(\mathbf{z}) = \mathbb{E}_{\mathbf{x}}\left[q_\phi(\mathbf{z}|\mathbf{x})\right]. \tag{3}$$

The main problem with the optimal prior in Eq. 3 is the summation over all $N$ training datapoints. Since $N$ could be very large (e.g., tens or hundreds of thousands), using such a prior is infeasible due to potentially very high memory demands. As an alternative approach, Tomczak & Welling (2018) proposed VampPrior, a new class of priors that approximate the optimal prior in the following manner:

$$p^*(\mathbf{z}) = \mathbb{E}_{\mathbf{x}} q_\phi(\mathbf{z}|\mathbf{x}) \approx \mathbb{E}_{r(\mathbf{u})} q_\phi(\mathbf{z}|\mathbf{u}), \tag{4}$$

where $\mathbf{u}$ is a *pseudoinput*, i.e., a variable mimicking real data, $r(\mathbf{u}) = \frac{1}{K}\sum_k \delta(\mathbf{u} - \mathbf{u}_k)$ is the distribution of $\mathbf{u}$ in the form of the mixture of Dirac's deltas, and $\{u_k\}_{k=1}^K$ are learnable parameters (we will refer to them as pseudoinputs as well). $K$ is a hyperparameter and is assumed to be smaller than the size of the training dataset, $K < N$. Pseudoinputs are randomly initialized before training and are learned along with model parameters by optimizing the ELBO objective using a gradient-based method.

In follow-up work, Egorov et al. (2021) suggested using a separate objective for pseudoinputs (a greedy boosting approach) and demonstrated the superior performance of such formulation in the continual learning setting. Here, we will present a different approximation to the optimal prior instead.

## 2.3 Ladder VAEs (a.k.a. Top-down VAEs)

We refer to models with many stochastic layers $L$ as deep hierarchical VAEs. They can differ in the way the prior and variational posterior distributions are factorized and parameterized. Here, we follow the factorization proposed in Ladder VAE (Sønderby et al., 2016) that considers the prior distribution over the latent variables factorized in an autoregressive manner:

$$p_\theta(\mathbf{z}_1, \ldots, \mathbf{z}_L) = p_\theta(\mathbf{z}_L) \prod_{l=1}^{L-1} p_\theta(\mathbf{z}_l | \mathbf{z}_{l+1:L}). \tag{5}$$

Next, using the top-down inference model results in the following variational posterior (Sønderby et al., 2016):

$$q_\phi(\mathbf{z}_1, \ldots, \mathbf{z}_L | \mathbf{x}) = q_\phi(\mathbf{z}_L | \mathbf{x}) \prod_{l=1}^{L-1} q_\phi(\mathbf{z}_l | \mathbf{z}_{l+1:L}, \mathbf{x}). \tag{6}$$

This factorization has been previously used by successful deep hierarchical VAEs, among others, NVAE (Vahdat & Kautz, 2020) and Very Deep VAE (VDVAE) (Child, 2021). It was shown empirically that such a formulation allows one to achieve state-of-the-art performance of the hierarchical VAEs on several image datasets.

## 3 Our model: Diffusion-based VampPrior VAE

In this work, we introduce the Diffusion-based VampPrior VAE (DVP-VAE). It is a deep hierarchical VAE model that approximates the optimal prior distribution at all levels of the hierarchical VAE in an efficient way.

### 3.1 VampPrior for hierarchical VAE

Directly using the VampPrior approximation (see Eq. 4) for deep hierarchical VAE can be very computationally expensive since it requires evaluating the variational posterior of all latent variables for $K$ pseudoinputs at each training iteration. Thus, (Tomczak & Welling, 2018) proposed a modification in which only the top latent variable uses VampPrior, namely:

$$p^*(\mathbf{z}_{1:L}) = \mathbb{E}_{\mathbf{x}} q_{\phi,\theta}(\mathbf{z}_{1:L} | \mathbf{x}) \approx \mathbb{E}_{\mathbf{x}} q_{\phi,\theta}(\mathbf{z}_L | \mathbf{x}) p_\theta(\mathbf{z}_{1:L-1}) \approx \mathbb{E}_{r(\mathbf{u})} q_{\phi,\theta}(\mathbf{z}_L | \mathbf{u}) p_\theta(\mathbf{z}_{1:L-1}), \tag{7}$$

where $r(\mathbf{u}) = \frac{1}{K} \sum_k \delta(\mathbf{u} - \mathbf{u}_k)$ with learnable pseudoinputs $\{u_k\}_{k=1}^K$. In this approach, there are a few problems: (i) how to pick the *best* number of pseudoinputs $K$, (ii) how to train pseudoinputs, and (iii) how to train the VampPrior in a scalable fashion. The last issue results from the first two problems and the fact that the dimensionality of pseudoinputs is the same as the original data, i.e., $\dim(\mathbf{u}) = \dim(\mathbf{x})$.

Here, we propose a different prior parameterization to overcome all these three problems. Our approach consists of three steps in which we approximate the VampPrior at all levels of the deep hierarchical VAE. We propose to *amortize* the distribution of pseudoinputs in VampPrior and use them to *directly* condition the prior distribution:

$$p^*(\mathbf{z}_{1:L}) = \mathbb{E}_{\mathbf{x}} q_{\phi,\theta}(\mathbf{z}_{1:L} | \mathbf{x}) \approx \mathbb{E}_{\mathbf{x}, r(\mathbf{u}|\mathbf{x})} p_\theta(\mathbf{z}_{1:L} | \mathbf{u}), \tag{8}$$

where we use $r(\mathbf{u}) = \mathbb{E}_{\mathbf{x}}[r(\mathbf{u}|\mathbf{x})]$. Using $r(\mathbf{u}|\mathbf{x})$, which is cheap to evaluate for any input, we avoid the expensive procedure of encoding pseudoinputs along with the inputs $\mathbf{x}$. Amortizing the VampPrior solves the problem of picking $K$ and helps with training pseudoinputs.

To define conditional distribution, we treat pseudoinputs $\mathbf{u}$ as a result of some noisy non-trainable transformation of the input datapoints $\mathbf{x}$. Let us consider a transformation $f : \mathcal{X}^D \to \mathcal{X}^P$, i.e., $\mathbf{u} = f(\mathbf{x}) + \sigma\varepsilon$, where $\varepsilon$ is a standard Gaussian random variable and $\sigma$ is the standard deviation. Note that by applying $f$, e.g., a linear transformation, we can lower the dimensionality of pseudoinputs, $\dim(\mathbf{u}) < \dim(\mathbf{x})$, resulting in better scalability. As a result, we get the following amortized distribution:

$$r(\mathbf{u}|\mathbf{x}) = \mathcal{N}(\mathbf{u}|f(\mathbf{x}), \sigma^2 I). \tag{9}$$

---

**Algorithm 1** $f_{\text{dct}}$: Create DCT-based pseudoinputs

---

**Input**: $\mathbf{x} \in \mathbb{R}^{c \times D \times D}, \mathbf{S} \in \mathbb{R}^{c \times d \times d}, d \in \mathbb{R}$
   $\mathbf{u}_{\text{DCT}} = \text{DCT}(\mathbf{x})$
   $\mathbf{u}_{\text{DCT}} = \text{Crop}(\mathbf{u}_{\text{DCT}}, d)$
   $\mathbf{u}_{\text{DCT}} = \frac{\mathbf{u}_{\text{DCT}}}{\mathbf{S}}$
**Return**: $\mathbf{u}_{\text{DCT}} \in \mathbb{R}^{c \times d \times d}$

---

**Algorithm 2** $f_{\text{dct}}^{\dagger}$: Invert DCT-based pseudoinputs

---

**Input**: $\mathbf{u}_{\text{DCT}} \in \mathbb{R}^{c \times d \times d}, \mathbf{S} \in \mathbb{R}^{c \times d \times d}, D$
   $\mathbf{u}_{\text{DCT}} = \mathbf{u}_{\text{DCT}} \cdot \mathbf{S}$
   $\mathbf{u}_{\text{DCT}} = \text{zero\_pad}(\mathbf{u}_{\text{DCT}}, D - d)$
   $\mathbf{u}_{\text{x}} = \text{iDCT}(\mathbf{u}_{\text{DCT}})$
**Return**: $\mathbf{u}_{\text{x}} \in \mathbb{R}^{c \times D \times D}$

---

The crucial part then is how to choose the transformation $f$. It is a non-trivial choice since we require the following properties of $f$: (i) it should result in $\dim(\mathbf{u}) < \dim(\mathbf{x})$, (ii) $\mathbf{u}$ should be a *reasonable* representation of $\mathbf{x}$, (iii) it should be *easily* computable (e.g., fast for scalability). We have two candidates for such transformations. First, we can consider a downsampled version of an image. Second, we propose to use a discrete cosine transform. We will discuss this approach in the following subsection.

Moreover, the outlined amortized VampPrior in the previous two steps seems to be a good candidate for efficient and scalable training. However, it does not seem to be suitable for generating new data. Therefore, we propose including pseudoinputs as the final level in our model and use a marginal distribution $\hat{r}(\mathbf{u})$ that approximates $r(\mathbf{u})$. Here, we propose to use a diffusion-based model for $\hat{r}(\mathbf{u})$.

### 3.2 DCT-based pseudoinputs

The first important component in our approach is the form of the non-trainable transformation from the input to the pseudoinput space. We assume that for $\mathbf{u}$ to be a *reasonable* representation of $\mathbf{x}$ means that $\mathbf{u}$ should preserve general patterns (information) of $\mathbf{x}$, but it does not necessarily contain any high-frequency details of $\mathbf{x}$. To achieve this, we propose to use a *discrete cosine transform*[1] (DCT) to convert the input into a frequency domain and then filter the high-frequency component.

**DCT** DCT (Ahmed et al., 1974) is a widely used transformation in signal processing for image, video, and audio data. For example, it is part of the JPEG standard (Pennebaker & Mitchell, 1992). For instance, consider a signal as a 3-dimensional tensor $\mathbf{x} \in \mathbb{R}^{c \times D \times D}$. DCT is a linear transformation that decomposes each channel $\mathbf{x}_i$ on a basis consisting of cosine functions of different frequencies: $\mathbf{u}_{DCT,i} = \mathbf{C}\mathbf{x}_i\mathbf{C}^{\top}$, where for all pairs $(k = 0, n)$: $\mathbf{C}_{k,n} = \sqrt{\frac{1}{D}}$, and for all pairs $(k, n)$ such that $k > 0$: $\mathbf{C}_{k,n} = \sqrt{\frac{2}{D}} \cos\left(\frac{\pi}{D}\left(n + \frac{1}{2}\right)k\right)$.

**Our transformation** We use the DCT transform as the first step in the procedure of computing pseudoinputs. Let us assume that each channel of $\mathbf{x}$ is $D \times D$. We select the desired size of the context $d < D$ and remove (crop) $D - d$ bottom rows and right-most columns for each channel in the frequency domain since they contain the highest-frequency information. Finally, we perform normalization using the matrix $\mathbf{S}$ that contains the maximal absolute value of each frequency. We calculate this matrix once (before training the model) using all the training data: $\mathbf{S} = \max_{\mathbf{x} \in \mathcal{D}_{\text{train}}} |\text{DCT}(\mathbf{x})|$. The complete procedure is described in Algorithm 1 and we denote it as $f_{\text{dct}}$.

In the frequency domain, a pseudoinput has a smaller spatial dimension than its corresponding datapoint. This allows us to use a small prior model and lower the memory consumption. However, we noticed empirically that conditioning the amortized VampPrior (see Eq. 8) on the pseudoinput in the original domain makes training easier. Therefore, the pseudo-inverse is applied as a part of the TopDown path. First, we start by multiplying the pseudoinput by the normalization matrix $\mathbf{S}$. Afterward, we pad each channel with zeros to account for the "lost" high frequencies. Lastly, we apply the inverse of the Discrete Cosine Transform (iDCT). We denote the procedure for converting a pseudoinput from the frequency domain to the data domain as $f_{\text{dct}}^{\dagger}$ and describe it in Algorithm 2.

---

[1]We consider the most widely used type-II DCT.

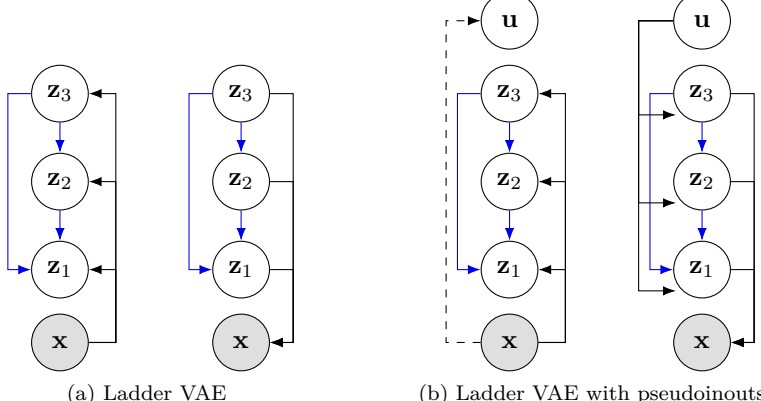

(a) Ladder VAE        (b) Ladder VAE with pseudoinouts

Figure 1: Graphical model of the TopDown hierarchical VAE with three latent variables (a) without pseudoinputs and (b) with pseudoinputs. The inference model (left) and the generative model (right) share parameters in the TopDown path (blue). The dashed arrow represents a non-trainable transformation.

### 3.3 Our Ladder VAE and Training Objective

We use the TopDown architecture and extend this model with a deterministic, non-trainable function to create the pseudoinput. In our generative model, pseudoinputs are treated as another set of latent variables:

$$p_\theta(\mathbf{x}, \mathbf{z}_{1:L}, \mathbf{u}) = p_\theta(\mathbf{x}|\mathbf{z}_{1:L})p_\theta(\mathbf{z}_{1:L}|\mathbf{u})r(\mathbf{u}), \tag{10}$$

$$p_\theta(\mathbf{z}_{1:L}|\mathbf{u}) = p_\theta(\mathbf{z}_L|\mathbf{u}) \prod_{l=1}^{L-1} p_\theta(\mathbf{z}_l|\mathbf{z}_{l+1:L}, \mathbf{u}). \tag{11}$$

Then, we choose variational posteriors in which pseudoinput latent variables are conditionally independent with all other latent variables given the datapoint $\mathbf{x}$, that is:

$$q_\phi(\mathbf{z}_{1:L}, \mathbf{u}|\mathbf{x}) = q_\phi(\mathbf{z}_{1:L}|\mathbf{x})r(\mathbf{u}|\mathbf{x}), \tag{12}$$

$$q_\phi(\mathbf{z}_{1:L}|\mathbf{x}) = q_\phi(\mathbf{z}_L|\mathbf{x}) \prod_{l=1}^{L-1} q_\phi(\mathbf{z}_l|\mathbf{z}_{l+1:L}, \mathbf{x}), \tag{13}$$

$$r(\mathbf{u}|\mathbf{x}) = \mathcal{N}(\mathbf{u}|f(\mathbf{x}), \sigma^2 I). \tag{14}$$

In the variational posteriors, we use the amortization of $r(\mathbf{u})$ outlined in Eq. 8.

Let us consider a Ladder VAE (a TopDown VAE) with three levels of latent variables. We depict the graphical model of this latent variable model in Figure 1a with the inference model on the left and the generative model on the right. Note that inference and generative models use shared parameters in the TopDown path, denoted by the blue arrows.

In Figure 1b we show a graphical model of our proposed model that additionally contains pseudoinputs $\mathbf{u}$. The generation model (Figure 1b right) is conditioned on the pseudoinput variable $\mathbf{u}$ at each level. This formulation is similar to the iVAE model (Khemakhem et al., 2020) in which the auxiliary random variable $\mathbf{u}$ is used to enforce the identifiability of the latent variable model. However, unlike iVAE, we do not treat $\mathbf{u}$ as a separate observed variable. Instead, we use a non-trainable transformation to obtain it during training and introduce the pseudoinput prior distribution $p_\gamma(\mathbf{u})$ with learnable parameters $\gamma$ to sample $\mathbf{u}$ at test time. This allows us to get unconditional samples from the model.

Finally, the Evidence Lower BOund for our model takes the following form:

$$\mathcal{L}(\mathbf{x}, \phi, \theta, \gamma) = \mathbb{E}_{q_\phi(\mathbf{z}_{1:L}|\mathbf{x})} \ln p_\theta(\mathbf{x}|\mathbf{z}_{1:L}) - \mathbb{E}_{q_\phi(\mathbf{u}|\mathbf{x})} D_{\mathrm{KL}} \left[ q_\phi(\mathbf{z}_{1:L}|\mathbf{x}) \| p_\theta(\mathbf{z}_{1:L}|\mathbf{u}) \right] - D_{\mathrm{KL}} \left[ r(\mathbf{u}|\mathbf{x}) \| r(\mathbf{u}) \right]. \tag{15}$$

The ELBO for our model gives a straightforward objective for training an approximate prior by minimizing the Kullback-Leibler between $r(\mathbf{u}|\mathbf{x})$ and the amortized VampPrior $r(\mathbf{u}) = \mathbb{E}_{\mathbf{x}}[r(\mathbf{u}|\mathbf{x})]$. However, calculating the KL-term with $r(\mathbf{u})$ is computationally demanding and, in fact, using $r(\mathbf{u})$ as the prior does not help us solve the original problem of using the VampPrior. Therefore, in the following section, we propose to use an approximation $\hat{r}_\gamma(\mathbf{u})$ instead.

### 3.4  Diffusion-based VampPrior

Even though pseudoinputs are assumed to be much simpler than the observed datapoint $\mathbf{x}$ (e.g., in terms of their dimensionality), a very flexible prior distribution $\hat{r}_\gamma(\mathbf{u})$ is required to ensure the high quality of the final samples. Since we cannot use the amortized VampPrior directly, following (Vahdat et al., 2021; Wehenkel & Louppe, 2021), we propose to use a diffusion-based generative model (Ho et al., 2020) as the prior and the approximation of $r(\mathbf{u})$.

Diffusion models are flexible generative models and, in addition, can be seen as latent variable generative models (Kingma et al., 2021). As a result, we have access to the lower bound on its log-likelihood function:

$$\log \hat{r}_\gamma(\mathbf{u}) \geq L_{vlb}(\mathbf{u}, \gamma) = \mathbb{E}_{q(\mathbf{y}_0|\mathbf{u})}[\ln r(\mathbf{u}|\mathbf{y}_0)] - D_{\mathrm{KL}}\left[q(\mathbf{y}_1|\mathbf{u})\|r(\mathbf{y}_1)\right] \tag{16}$$
$$- \sum_{i=1}^{T} \mathbb{E}_{q(\mathbf{y}_{i/T}|\mathbf{u})} D_{\mathrm{KL}}\left[q(\mathbf{y}_{(i-1)/T}|\mathbf{y}_{i/T}, \mathbf{u})\|r_\gamma(\mathbf{y}_{(i-1)/T}|\mathbf{y}_{i/T})\right].$$

We provide more details on diffusion models and the derivation of the ELBO in Appendix A. We refer to this prior as *Diffusion-based VampPrior* (DVP). This prior allows us to sample infinitely many pseudoinputs, unlike the original VampPrior that uses a fixed set of $K$ pseudoinputs.

Now, we can plug this lower bound into the objective (Eq. 15) and obtain the final objective of our Ladder VAE with pseudoinputs and the Diffusion-based VampPrior (dubbed DVP-VAE):

$$\max_{\theta, \phi, \gamma} \mathbb{E}_\mathbf{x}\left[\mathbb{E}_{q_\phi(\mathbf{z}|\mathbf{x})} \ln p_\theta(\mathbf{x}|\mathbf{z}) - \mathbb{E}_{r(\mathbf{u}|\mathbf{x})} D_{\mathrm{KL}}\left[q_\phi(\mathbf{z}|\mathbf{x})\|p_\theta(\mathbf{z}|\mathbf{u})\right] + \mathbb{H}[r(\mathbf{u}|\mathbf{x})] - \mathbb{E}_{r(\mathbf{u}|\mathbf{x})} L_{vlb}(\mathbf{u}, \gamma)\right]. \tag{17}$$

Note that $\mathbb{H}[r(\mathbf{u}|\mathbf{x})] = \frac{P}{2}\log(2\pi \exp \sigma^2)$ where $\sigma$ is a learnable parameter (see Eq. 9) and $P$ is the dimensionality of the pseudoinput.

## 4  Model Details: Architecture and parameterization

The model architecture and parameterization are crucial to the scalability of the model. In this section, we discuss the specific choices we made. The starting point for our architecture is the architecture proposed in VDVAE (Child, 2021). However, there are certain differences. We schematically depict our architecture in Figure 2a. We consider a hierarchical TopDown VAE with $L$ stochastic layers, namely, latent variables $\mathbf{z}_1, \ldots, \mathbf{z}_L$. We assume that each latent variable has the same number of channels, but they differ in spatial dimensions: $\mathbf{z}_l \in \mathbb{R}^{c \times h_l \times w_l}$. We refer to different spatial dimensions of the latent space as *scales*.

### 4.1  Bottom-up

The bottom-up part corresponds to calculations of intermediary variables dependent on $\mathbf{x}$. We follow the implementation of Child (2021) for it. We start from the bottom-up path depicted in Figure 2a (left), which is fully deterministic and consists of several ResNet blocks (see Figure 2c). The input is processed by $N_{\mathrm{enc}}$ blocks at each scale, and the output of the last resnet block of each scale is passed to the TopDown path in Figure 2a (right). Note that here $N_{\mathrm{enc}}$ is a separate hyperparameter that does not depend on the number of stochastic layers $L$.

### 4.2  TopDown

The TopDown path depicted in Figure 2a (right) computes the parameters of the variational posterior and the prior distribution starting from the top latent variable $\mathbf{z}_L$.

The first step is the pseudoinput block shown in Figure 2d. Using the deterministic function $f_{\mathrm{dct}}$, it creates the pseudoinput random variable from the input $\mathbf{x}$ (see Algorithm 1) that is used to train the Diffusion-based VampPrior $\hat{r}_\gamma(\mathbf{u})$. At test time, a pseudoinput is sampled using this unconditional prior. The pseudoinput sample is then converted back to the input domain (see Algorithm 2) and used to condition prior distributions at all levels $p_\theta(\mathbf{z}_{1:L}|\mathbf{u})$.

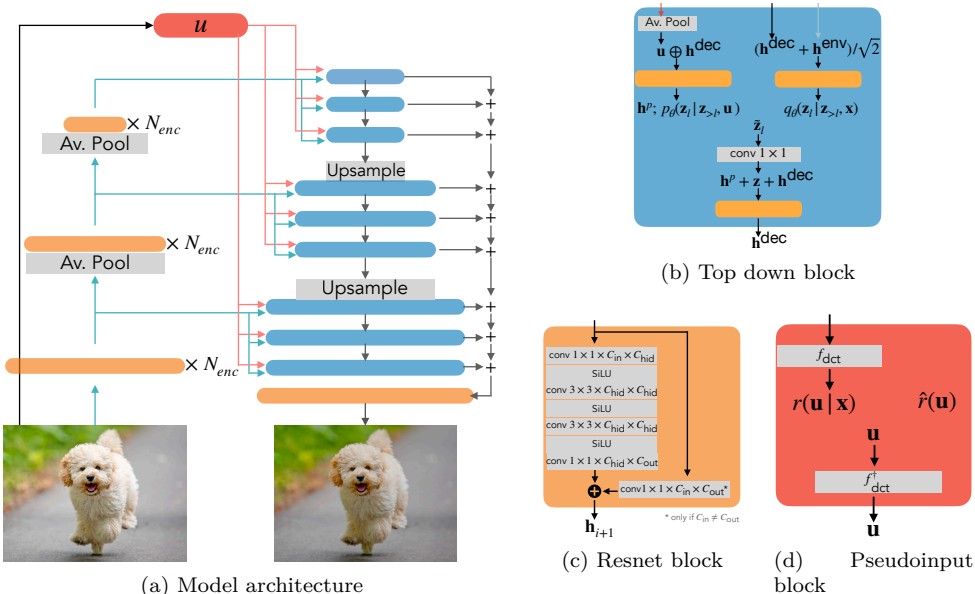

Figure 2: A diagram of the DVP-VAE: TopDown hierarchical VAE with the diffusion-based VampPrior. (a) A *BottomUp* path (left) and a *TopDown* path (right). (b) A *TopDown* block that takes features from the block above $\mathbf{h}^{dec}$, encoder features $\mathbf{h}^{enc}$ (only during training) and a pseudoinput $\mathbf{u}$ as inputs. (c) A single Resnet block. (d) A single pseudoinput block.

Next, the model has $L$ TopDown blocks depicted in Figure 2b. Each TopDown block takes deterministic features from the corresponding scale of the bottom-up path denoted as $h_{enc}$, the output of the pseudoinput block $\mathbf{u}$, and deterministic features from the TopDown block above $h_{dec}$ as inputs. Our implementation of this block is similar to the VDVAE architecture, but there are several differences that we summarize below:

- *Incorporating pseudoinputs*
  We concatenate the pseudoinput (properly reshaped using average pooling) with $h_{dec}$ to compute the parameters of the prior distribution.

- *Variational posterior parameters*
  We assume that both $h_{dec}$ and $h_{enc}$ have the same number of channels, allowing us to sum them instead of concatenating. This reduces the total number of parameters and the memory consumption.

- *Additional ResNet Connections*
  Our TopDown block has three ResNet blocks (Figure 2c). In contrast to our architecture, in VDVAE only the block that updates $h_{dec}$ has a residual connection.

We did not observe any training instabilities and did not apply the *gradient skipping* used in Child (2021).

## 4.3 Latent Aggregation in Conditional Likelihood

The last important element of our architecture is the *aggregation of latents*. Let us denote samples from either the variational posteriors $q_\phi(\mathbf{z}_{1:L}|\mathbf{x})$ during training or the prior $p_\theta(\mathbf{z}_{1:L}|\mathbf{u})$ during generating new data as $\tilde{\mathbf{z}}_1, \ldots, \tilde{\mathbf{z}}_L$. Furthermore, let $\boldsymbol{h}_1$ be the output of the last TopDown block. These deterministic features are computed as a function of all samples $\tilde{\mathbf{z}}_1, \ldots, \tilde{\mathbf{z}}_L$. Therefore, it can be used to calculate the final likelihood value. However, we observe empirically that in such parameterization some layers of latent variables tend to be completely ignored by the model. Instead, we propose to enforce a strong connection between the conditional likelihood and all latent variables by explicitly conditioning on all of the sampled latent variables, namely:

$$p_\theta(\mathbf{x}|\mathbf{z}_{1:L}) = p_\theta\big(\mathbf{x}\big|\mathrm{NN}\big(\frac{1}{\sqrt{L}}\sum_l \tilde{\mathbf{z}}_l\big)\big). \tag{18}$$

We refer to this as the *latent aggregation*. We show empirically in the experimental section that this leads to a consistently high ratio of active units.

## 5  Related Work

**Latent prior in VAEs**  The original VAE formulation uses the standard Gaussian distribution as a prior over the latent variables. This can be an overly simplistic choice, as the prior minimizing the Evidence Lower bound is given by the aggregated posterior (Hoffman & Johnson, 2016; Tomczak & Welling, 2018). Furthermore, using a unimodal prior with multimodal real-world data can lead to non-smooth encoder or meaningless distances in the latent space Bozkurt et al. (2019). More flexible prior distributions proposed in the literature include the Gaussian Mixture Model (Jiang et al., 2016; Nalisnick et al., 2016; Tran et al., 2022), the autoregressive normalizing flow (Chen et al., 2017), the autoregressive model (Gulrajani et al., 2016; Sadeghi et al., 2019), the rejection sampling distribution with the learned acceptance function (Bauer & Mnih, 2019), the diffusion-based prior (Vahdat et al., 2021; Wehenkel & Louppe, 2021). The VampPrior (Tomczak & Welling, 2018) proposes using an approximation of the aggregated posterior as a prior distribution. The approximation is constructed using learnable pseudoinputs to the encoder. This work can be seen as an efficient extension of the VampPrior to deep hierarchical VAE, which also utilizes a diffusion-based prior over the pseudoinputs.

**Auxiliary Variables in VAEs**  Several works consider auxiliary variables $\mathbf{u}$ as a way to improve the flexibility of the variational posterior. Maaløe et al. (2016) use auxiliary variables with one-level VAE to improve the variational approximation while keeping the generative model unchanged. Salimans et al. (2015) use Markov transition kernel for the same expressivity purpose. The authors treat intermediate MCMC samples as an auxiliary random variable and derive evidence lower bound of the extended model. Ranganath et al. (2016) introduce hierarchical variational models. They increase the flexibility of the variational approximation by imposing prior on its parameters. In this setting, it assumes that the latent variable $\mathbf{z}$ and the auxiliary variable $\mathbf{u}$ are not conditionally independent and the variational posterior factorizes, for example, as follows:

$$q_\phi(\mathbf{u}, \mathbf{z}|\mathbf{x}) = q_\phi(\mathbf{u}|\mathbf{x})q_\phi(\mathbf{z}|\mathbf{u}, \mathbf{x}). \tag{19}$$

In this work, in contrast, we use auxiliary variable to increase the prior flexibility and use conditional independence assumption in the variational posterior:

$$q_\phi(\mathbf{u}, \mathbf{z}|\mathbf{x}) = q_\phi(\mathbf{u}|\mathbf{x})q_\phi(\mathbf{z}|\mathbf{x}). \tag{20}$$

Khemakhem et al. (2020) consider the non-identifiability problem of VAEs. They propose to use auxiliary observation $\mathbf{u}$ and use it to condition the prior distribution. This additional observation is similar to the pseudoinputs that we consider in our work. However, we define a way to construct $\mathbf{u}$ from the input and learn a prior distribution to sample it during inference, while Khemakhem et al. (2020) require $\mathbf{u}$ to be observed both during training and at the inference time.

Similarly to our work, (Klushyn et al., 2019) consider hierarchical prior $p_\theta(\mathbf{z}|\mathbf{u})p(\mathbf{u})$. However, they treat $\mathbf{u}$ rather as a second layer of latent variables and learn a variational posterior in the form $q_\phi(\mathbf{u}, \mathbf{z}|\mathbf{x}) = q_\phi(\mathbf{u}|\mathbf{z})q_\phi(\mathbf{z}|\mathbf{x})$.

**Latent Variables Aggregation**  There are different ways in which the conditional likelihood $p_\theta(\mathbf{x}|\mathbf{z}_{1:L})$ can be parameterized. In LadderVAE (Sønderby et al., 2016), where TopDown hierarchical VAE was originally proposed, the following formulation is used:

$$p_\theta(\mathbf{x}|\mathbf{z}_{1:L}) = p_\theta(\mathbf{x}|\text{NN}(\mathbf{z}_1)). \tag{21}$$

That is, the conditional likelihood depends directly on the bottom latent variable $\mathbf{z}_1$ only.

Later, NVAE (Vahdat & Kautz, 2020) and VDVAE (Child, 2021) use a deterministic path in the TopDown architecture in the conditional likelihood, namely:

$$p_\theta(\mathbf{x}|\mathbf{z}_{1:L}) = p_\theta(\mathbf{x}|\text{NN}(\mathbf{h}_1)). \tag{22}$$

Table 1: The test performance: negative log likelihood on MNIST and OMNIGLOT, and bits-per-dimension (BPD) on CIFAR10.
‡ Results with data augmentation. Standard deviation reported in parentheses.
* Results averaged over 4 random seeds.

| Model | L | MNIST | OMNIGLOT | CIFAR10 | | |
|---|---|---|---|---|---|---|
| | | $-\log p(\mathbf{x}) \leq\ \downarrow$ | | Size | L | BPD $\leq\ \downarrow$ |
| **DVP-VAE** (ours) | 8 | **77.10**\*(0.05) | **89.07**\*(0.10) | 20M | 28 | **2.73** |
| Attentive VAE (Apostolopoulou et al., 2022) | 15 | 77.63 | 89.50 | 119M | 16 | 2.79 |
| CR-NVAE (Sinha & Dieng, 2021) | 15 | 76.93‡ | — | 131M | 30 | 2.51‡ |
| VDVAE (Child, 2021) | — | — | — | 39M | 45 | 2.87 |
| OU-VAE (Pervez & Gavves, 2021) | 5 | 81.10 | 96.08 | 10M | 3 | 3.39 |
| NVAE (Vahdat & Kautz, 2020) | 15 | 78.01 | — | — | 30 | 2.91 |
| BIVA(Maaløe et al., 2019) | 6 | 78.41 | 91.34 | 103M | 15 | 3.08 |
| VampPrior (Tomczak & Welling, 2018) | 2 | 78.45 | 89.76 | — | — | — |
| LVAE (Sønderby et al., 2016) | 5 | 81.74 | 102.11 | — | — | — |
| IAF-VAE(Kingma et al., 2016) | — | 79.10 | — | — | 12 | 3.11 |

Note that deterministic features depend on all the latent variables. However, we propose to use a more explicit dependency on latent variables in Eq. 18. Our idea bears some similarities with Skip-VAE (Dieng et al., 2019). Skip-VAE proposes to add a latent variable to each layer of the neural network parameterizing decoder of the VAE with a single stochastic layer. In this work, instead, we add all the latent variables together to parameterize conditional likelihood.

# 6 Experiments

## 6.1 Settings

We evaluate DVP-VAE on dynamically binarized MNIST (LeCun, 1998) and OMNIGLOT (Lake et al., 2015). Furthermore, we conduct experiments on natural images using the CIFAR10 dataset (Alex, 2009). We provide all the hyperparameters for training DVP-VAE in Appendix C.1 and in the code repository[2].

## 6.2 Main Quantitative and Qualitative Results

We report all results in Table 1, where we compare the proposed approach with other hierarchical VAEs. We observe that DVP-VAE outperforms most of the VAE models[3] while using fewer parameters than other models. For instance, on CIFAR10, our DVP-VAE requires 20M weights to beat Attentive VAE with about 6 times more weights. Furthermore, because of the smaller model size, we were able to obtain all the results using a single GPU. We show the unconditional samples in Figure 3 (see Appendix D for more samples). The top row of each image shows samples from the diffusion-based VampPrior (i.e., pseudoinputs), while the second row shows corresponding samples from the VAE. We observe that, as expected, pseudoinputs define the general appearance of an image, while a lot of details are added later by the TopDown decoder. This effect can be further observed in Figure 4 where we plot the reconstructions using different numbers of latent variables. In the first row, only a pseudoinput corresponding to the original image is used (i.e., $\mathbf{u} \sim r(\mathbf{u}|\mathbf{x})$) while the remaining latent variables are sampled from the prior with low temperature. Each row below uses more latent variables from the variational posterior grouped by the scales. Namely, the second row uses the pseudoinput above and all the $4 \times 4$ latent variables from the variational posterior, then the third row uses additionally $8 \times 8$ latent variables, and so on.

## 6.3 Ablation Studies

**Training stability and convergence** In all of our experiments, we did not observe many training instabilities. Unlike many contemporary VAEs, we did not use gradient skipping (Child, 2021), spectral

---

[2]https://github.com/AKuzina/dvp_vae
[3]CR-NVAE (Sinha & Dieng, 2021) uses NVAE model and considerably improves its performance by applying data augmentations. We did not use any augmentations to compare fairly with all other hierarchical VAEs and were able to get a better NLL than NVAE and more recent hierarchical VAE models.

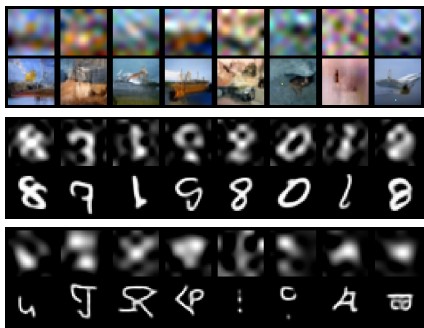 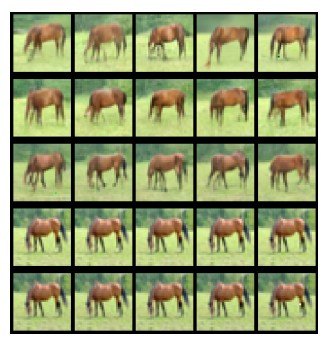 

Figure 3: Unconditional samples from the Diffusion-based VampPrior (top) and corresponding samples from the DVP-VAE (bottom).

Figure 4: *Generative reconstruction*s. The top row is using a pseudoinput sampled from $r(\mathbf{u}|\mathbf{x})$ **only**.

Table 2: Test performance for the model trained for 500 epochs (or approximately 200k training iterations) on CIFAR10.

| L | Size | BPD$\leq \downarrow$ | AU $\uparrow$ |
|---|---|---|---|
| 20 | 24M | 2.99 | 94% |
| 28 | 32M | 2.94 | 93% |
| 36 | 40M | 2.89 | 98% |
| 44 | 48M | 2.84 | 97% |

Table 3: Training settings for the model trained on CIFAR-10 compared to two VDVAE implementations.

| | VDVAE (Child, 2021) | Efficient-VDVAE (Hazami et al., 2022) | DVP-VAE (ours) |
|---|---|---|---|
| L | 45 | 47 | 45 |
| Size | 39M | 57M | 38M |
| Optimizer | AdamW | Adamax | Adamax |
| Learning rate | 2e-4 | 1e-3 | 1e-3 |
| Grad. smoothing | – | Yes | – |
| Grad. skip | 400 | 800 | – |
| Training iter. | 1.1M | 0.8M | **0.4M** |
| Test BPD | 2.87 | 2.87 | 2.86 |

normalization (Vahdat & Kautz, 2020), or softmax parameterization of variances (Hazami et al., 2022). We use the Adamax version of the Adam optimizer (Kingma & Ba, 2015) following Hazami et al. (2022), as it demonstrated much better convergence for the model with a mixture of discretized logistic likelihood.

First, we observe a consistent performance improvement as we increase the model size and the number of stochastic layers. In Table 2, we report test performance and the percentage of active units (see Section 6.3 for details) for models of different stochastic depths trained on the CIFAR10 dataset. We train each model for 500 epochs, which corresponds to less than 200k training iterations. Additionally, we report gradient norms and training and validation losses for all four models in Appendix B.

To demonstrate the advantage of the proposed architecture, we compare our model to a closest deep hierarchical VAE architecture: Very Deep VAE(Child, 2021). For this experiment, we chose hyperparameters closest to Table 4 in Child (2021) (CIFAR-10). That is, our model has 45 stochastic layers and a comparable number of trainable parameters. Furthermore, following Child (2021), we train this model with a batch size of 32, a gradient clipping threshold of 200, and an EMA rate of 0.9998. However, in DVP-VAE, we were able to eliminate gradient skipping and gradient smoothing. We report the difference in key hyperparameters and test performance in Table 3. We also add a comparison with Efficient-VDVAE (see Table 3 in Hazami et al. (2022)). We observe that DVP-VAE achieves comparable performance within much fewer training iterations than both VDVAE implementations.

**Latent Aggregation Increases Latent Space Utilization** Next, we test the claim that latent variable aggregation discussed in Sec. 4.3 improves latent space utilization. We use Active Units (AU) metric (Burda et al., 2015), which can be calculated for a given threshold $\delta$ as follows:

$$AU = \frac{\sum_{l=1}^{L} \sum_{i=1}^{M_l} [A_{l,i} > \delta]}{\sum_{l=1}^{L} M_l}, \tag{23}$$

$$\text{where } A_l = \text{Var}_{q^{\text{test}}(\mathbf{x})} \mathbb{E}_{q_\phi(\mathbf{z}_{l+1:L}|\mathbf{x})} \mathbb{E}_{q_\phi(\mathbf{z}_l|\mathbf{z}_{l+1:L},\mathbf{x})} [\mathbf{z}_l]. \tag{24}$$

Table 4: Active Units for the DCT-VAE with and without latent aggregation.

| Latent Aggr. | Size | L | AU ↑ |
|---|---|---|---|
| MNIST | | | |
| ✗ | 0.7M | 8 | 33.2% |
| ✓ | 0.7M | 8 | 91.5% |
| OMNIGLOT | | | |
| ✗ | 1.3M | 8 | 71.3% |
| ✓ | 1.3M | 8 | 93.4% |
| CIFAR10 | | | |
| ✓ | 19.5M | 28 | 98% |

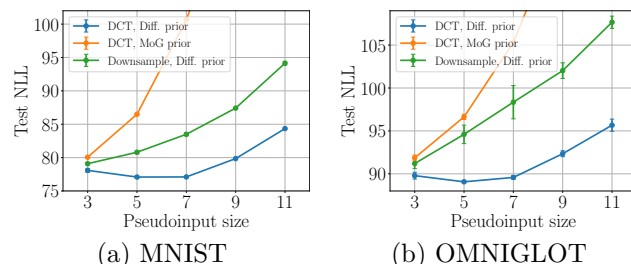

(a) MNIST    (b) OMNIGLOT

Figure 5: Ablation study of for the pseudoinputs type (DCT and Downsampled image), pseudoinputs prior (Diffusion model and Mixture of Gaussians) and pseudoinputs size (ranging from $3 \times 3$ to $11 \times 11$). Each configuration is trained with four different random seeds.

Table 5: Test NLL for the model with and without pseudoinputs. Averaged over 4 random seeds, standard deviation in parenthesis.

| Pseudoinputs | Size | NLL↓ |
|---|---|---|
| MNIST | | |
| ✗ | 0.6M | 78.85 (0.24) |
| ✓ | 0.7M | 77.10 (0.05) |
| OMNIGLOT | | |
| ✗ | 1.1M | 89.52 (0.23) |
| ✓ | 1.3M | 89.07 (0.10) |

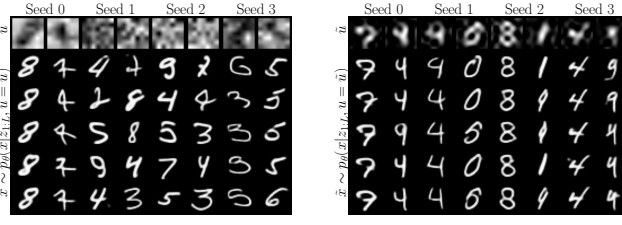

(a) Learnable pseudoinputs (b) DCT pseudoinputs

Figure 6: Samples from the pseudoinputs prior $\tilde{\mathbf{u}} \sim \hat{r}(\mathbf{u})$ (top row) and corresponding samples from the model $\tilde{\mathbf{x}} \sim p_\theta(\mathbf{x}|\mathbf{z}_{1:L}, \mathbf{u} = \tilde{\mathbf{u}})$ (other rows). Columns corresponds to models trained with different random seeds.

Here $M_l$ is the dimensionality of the stochastic layer $l$, $[B]$ is the Iverson bracket, which equals 1 if $B$ is true and 0 otherwise, and Var stands for variance. Following Burda et al. (2015), we use the threshold $\delta = 0.01$. The higher the share of active units, the more efficient the model is in utilizing its latent space.

We report results in Table 4 and observe that the model with latent aggregation always attains more than 90% of active units. Furthermore, latent aggregation considerably improves the utilization of the latent space if we compare it with exactly the same model but with the conditional likelihood parameterized using deterministic feature from the TopDown path (see Eq. 22).

**Amortized VampPrior Improves BPD** Further, we test how the proposed amortized VampPrior improves model performance as measured by the negative log-likelihood. We report results in Table 5 and observe that DVP-VAE always has a better NLL metric compared to the deep hierarchical VAE with the same architecture and number of stochastic layers. Due to additional diffusion-based prior over pseudoinputs, DVP-VAE has slightly more trainable parameters. However, because of the small spatial dimensionality of the pseudoinputs, we were able to keep the size of the two models comparable.

**Pseudoinputs type, size and prior** We conduct an extensive ablation study regarding pseudoinputs. First, we train VAE with two types of pseudoinputs: DCT and Downsampled images. Moreover, we vary the spatial dimensions of the pseudoinputs between $3 \times 3$ and $11 \times 11$. We expect that a smaller pseudoinputs size will be an easier task for the prior $\hat{r}(\mathbf{u})$, but will constitute a poorer approximation of an optimal prior. The larger pseudoinput size, on the other hand, results in a better optimal prior approximation since more information about the datapoint $\mathbf{x}$ is preserved. However, it becomes harder for the prior to achieve good results since we keep the prior model size fixed. In Figure 5 we observe that the DCT-based transformation performs consistently better across various sizes and datasets.

**Trainable Pseudoinputs** In the VampPrior, the optimal prior is approximated using learnable pseudoinputs. In this work, on the other hand, we propose to use fixed linear transformation instead. To further verify whether a fixed transformation like DCT is reasonable, we checked a learnable linear transformation. We present in Figure 6 that the learnable linear transformation of the input exhibits unstable behavior in terms of the **quality** of learned pseudoinput. The top row of Figure 6(a) shows samples from the trained prior and the corresponding samples from the decoder. We observe that only one out of four models with learnable pseudoinputs was able to learn a visually meaningful representation of the data (seed 0), which also resulted in very high variance of the results (rows below). For other models (e.g., Seed 1 and Seed 2), the same pseudoinput sample corresponds to completely different datapoints.

This lack of consistency motivates us to use a non-trainable transformation for obtaining pseudoinputs. In Figure 6 (b), we show the expected behavior of sampling semantically meaningful pseudoinputs that is consistent across random seeds.

Table 6: Wall-clock time, memory consumption and sampling time for DVP and VampPrior.

| | K | TRAIN TIME | GPU MEM. | SIZE | SAMPLE TIME |
|---|---|---|---|---|---|
| | | MNIST (32 CHANNELS) | | | |
| DVP | N/A | 45s | 3Gb | 0.7M | 2.2 |
| Vamp | 500 | 45s | 4Gb | 1.0M | 0.2 |
| | 1000 | 60s | 6Gb | 1.4M | 0.3 |
| | | MNIST (64 CHANNELS) | | | |
| DVP | N/A | 50s | 4Gb | 2.4M | 2.3 |
| Vamp | 500 | 65s | 7Gb | 2.7M | 0.5 |
| | 1000 | 90s | 10Gb | 3.1M | 0.8 |
| | | CIFAR10 | | | |
| DVP | N/A | 370s | 12Gb | 19.5M | 4.0 |
| | 500 | 430s | 29Gb | 20.5M | 1.5 |
| Vamp | 750 | 500s | 38Gb | 21.3M | 1.8 |
| | 1000 | OOM | >40Gb | 22.1M | — |

**Scalability** Here, we study how DVP-VAE scales as we increase model size and input size in comparison with VampPrior. For this, we implement VampPrior as proposed by Tomczak & Welling (2018), where the top latent variable is trained with the VampPrior and the other layers with the conditional Gaussian (see Eq. 7). We use the same architecture as in main experiments for MNIST (32 channels) and CIFAR10 (see Table 7). Additionally, we train model with doubled number of channels on MNIST (64 channels).

In Table 6, we report the training time (second per epoch), GPU memory utilization and total number of trainable parameters. We observe that Vamp-Prior almost always utilizes significantly more memory and requires longer training time. The difference is less visible on a small model and small input size (MNIST, 32 channels). However, as we double number of channels, both training time and memory utilization for VampPrior grows much faster. As a result, for a bigger input size (CIFAR10 dataset) a model with more than 750 pseudoinputs does not fit into a single A100 GPU. For 500 pseudoinputs, memory utilization is already 2.5 times higher for VampPrior compared to DVP-VAE.

## 7 Limitations

One limitation of DVP-VAE is the long sampling time reported in Table 6 (sec. per 1000 images). It is a direct consequence of using a diffusion-based prior. While diffusion is a powerful generative model that allows us to achieve outstanding performance, it is known to be slow at sampling. In our experiments, we use 50 diffusion steps to generate a pseudoinput, however, there are efficient distillation techniques (Salimans & Ho, 2022; Geng et al., 2024) that can be applied to mitigate this issue and reduce the number of diffusion steps to just a single forward pass through the model. We leave this optimization for future work.

Furthermore, within DVP-VAE, we add an extra learnable block to the model, namely, the diffusion-based prior over pseudoinputs, as a result, additional modelling choices should be made. We provide an ablation study to show the effect of these choices on the performance of the model.

## 8    Conclusion

In this work, we introduce DVP-VAE, a new class of deep hierarchical VAEs with the diffusion-based Vamp-Prior. We propose to use a VampPrior approximation which allows us to use it with hierarchical VAEs with little computational overhead. We show that the proposed approach demonstrate competitive performance in terms of the negative log-likelihood on three benchmark datasets with much fewer parameters and stochastic layers compared to the best performing contemporary hierarchical VAEs.

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

# A  Diffusion probabilistic models

Diffusion Probabilistic Models or Diffusion-based Deep Generative Models (Ho et al., 2020; Sohl-Dickstein et al., 2015) constitute a class of generative models that can be viewed as a special case of the Hierarchical VAEs (Huang et al., 2021; Kingma et al., 2021; Tomczak, 2022; Tzen & Raginsky, 2019). Here, we follow the definition of the variational diffusion model (Kingma et al., 2021). We use the diffusion model as a prior over the pseudoinputs $\mathbf{u}$.

**Forward diffusion process**

The *forward* diffusion *process* runs forward in time and gradually adds noise to the input $\mathbf{u}$ as follows:

$$q(\mathbf{y}_t|\mathbf{u}) = \mathcal{N}(\mathbf{y}_t; \alpha_t\mathbf{u}, (1 - \alpha_t^2)\mathbf{I}). \tag{25}$$

where $\mathbf{y}_t$ are auxiliary latent variables indexed by time $t \in [0, 1]$, $\alpha_t$ is chosen in such a way that the signal-to-noise ratio decreases monotonically over time.

Since the conditionals in the forward diffusion can be seen as Gaussian linear models, we can analytically calculate the following distributions for $t > s$:

$$q(\mathbf{y}_t|\mathbf{y}_s, \mathbf{u}) = \mathcal{N}(\mathbf{y}_{t-1}; \tilde{\mu}(\mathbf{y}_t, \mathbf{u}), \tilde{\sigma}(t, s)\mathbf{I}), \tag{26}$$

$$\text{where } \tilde{\mu}(\mathbf{y}_t, \mathbf{u}) = \frac{\alpha_t\left(1 - \alpha_s^2\right)}{\alpha_s\left(1 - \alpha_t^2\right)}\mathbf{y}_t + \frac{\alpha_s^2 - \alpha_t^2}{(1 - \alpha_t^2)\alpha_s}\mathbf{u}, \tag{27}$$

$$\tilde{\sigma}(t, s) = \frac{\left(\alpha_s^2 - \alpha_t^2\right)}{\alpha_s^2}\frac{\left(1 - \alpha_s^2\right)}{\left(1 - \alpha_t^2\right)}. \tag{28}$$

**Backward diffusion process**

Similarly, we define a generative model, also referred to as the *backward* (or *reverse*) *process*, as a Markov chain with Gaussian transitions starting with $r(\mathbf{y}_1) = \mathcal{N}(\mathbf{y}_1|\mathbf{0}, \mathbf{I})$. We discretize time uniformly into $T$ timestamps of length $1/T$:

$$r(\mathbf{y}_0, \ldots, \mathbf{y}_1) = r(\mathbf{y}_1)\prod_{i=1}^{T} r_\gamma(\mathbf{y}_{(i-1)/T}|\mathbf{y}_{i/T}), \tag{29}$$

where $r_\gamma(\mathbf{y}_{t-1}|\mathbf{y}_t) = \mathcal{N}(\mathbf{y}_{t-1}; \mu_\gamma(\mathbf{y}_t, t), \Sigma_\gamma(\mathbf{y}_t, t))$.

**The Likelihood term**

Common practice is to define the likelihood term as being proportional to the first step of the forward process: $r(\mathbf{u}|\mathbf{y}_0) \propto q(\mathbf{y}_0|\mathbf{u})$. Since we assume that the pseudoinput random variable $\mathbf{u}$ is continuous, we get the Gaussian likelihood distribution:

$$r(\mathbf{u}|\mathbf{y}_0) = \mathcal{N}(\mathbf{u}|\mathbf{y}_0/\alpha_0, \sigma_0^2/\alpha_0^2 I) \tag{30}$$

Note that the same likelihood term was used for continuous atom positions in the equivariant diffusion model (Hoogeboom et al., 2022).

**Training objective**

We can use (25) and (26) to define the variational lower bound as follows:

$$\hat{r}_\gamma(\mathbf{u}) \geq L_{vlb}(\mathbf{u}, \gamma) = \underbrace{\mathbb{E}_{q(\mathbf{y}_0|\mathbf{u})}[\ln r(\mathbf{u}|\mathbf{y}_0)]}_{-L_0} - \underbrace{D_{\mathrm{KL}}\left[q(\mathbf{y}_1|\mathbf{u})\|r(\mathbf{y}_1)\right]}_{L_1} \tag{31}$$

$$- \underbrace{\sum_{i=1}^{T} \mathbb{E}_{q(\mathbf{y}_{i/T}|\mathbf{u})} D_{\mathrm{KL}}\left[q(\mathbf{y}_{(i-1)/T}|\mathbf{y}_{i/T}, \mathbf{u})\|r_\gamma(\mathbf{y}_{(i-1)/T}|\mathbf{y}_{i/T})\right]}_{L_T}.$$

Here we refer to $L_0$ as the reconstruction loss, $L_1$ as the prior loss, and $L_T$ as the diffusion loss with $T$ steps.

# B    Training Stability: depth

We report the $\ell_2$-norm of the gradient for each iteration training for models of different stochastic depth in Figure 7. We observe very high gradient norms for the first few training iterations but no spikes at later stages of training. This happens because we did not initiate the KL-term to be zero at the beginning of the training and it tends to be large at the initialization step for very deep VAEs. However, we did not observe our model diverge since the gradient is clipped to reasonable values (200 in these experiments) and after the first few gradient updates, the KL-term goes to reasonable values. Moreover, we plot training and validation losses in Figure 8.

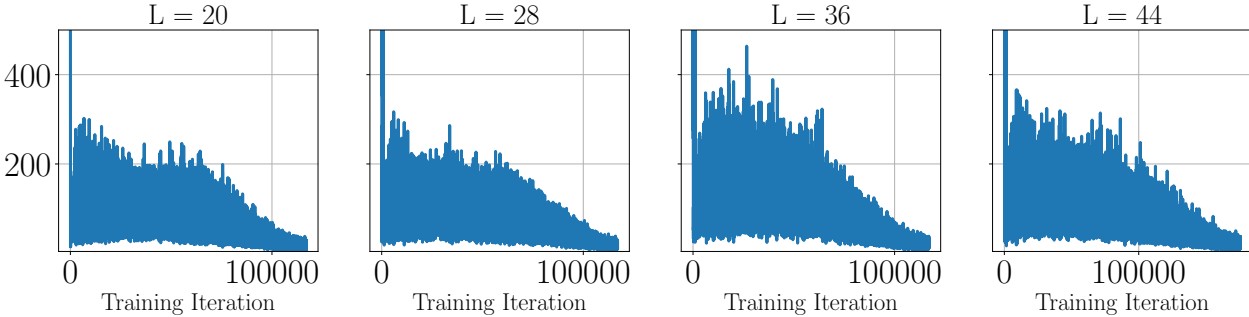

Figure 7: The gradient norm at each training iteration. Models were trained on CIFAR10 with different stochastic depths L.

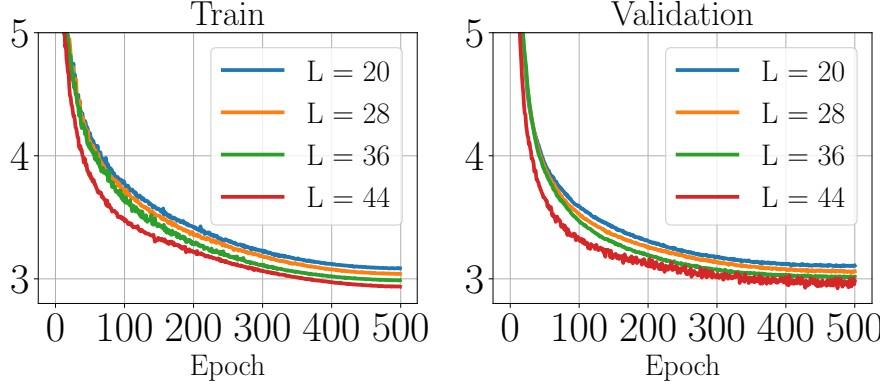

Figure 8: The ELBO per pixel on train (left) and validation (right) dataset. Models were trained on CIFAR10 with different stochastic depths L.

## C   Model details

### C.1   Hyperparameters

In Table 7, we report all hyperparameter values that were used to train the DVP-VAE.

**The pseudoinput prior**   We use the diffusion generative model as a prior over pseudoinputs. As a backbone, we use the UNet implementation from (Dhariwal & Nichol, 2021), available on GitHub[4] with the hyperparameters provided in Table 7.

Table 7: Full list of hyperparameters.

| | | MNIST | OMNIGLOT | CIFAR10 |
|---|---|---|---|---|
| Optimization | # Epochs | 300 | 500 | 3000 |
| | Batch Size (per GPU) | 250 | 250 | 128 |
| | # GPUs | 1 | 1 | 1 |
| | Optimizer | Adamax | Adamax | Adamax |
| | Scheduler | Cosine | Cosine | Cosine |
| | Starting LR | 1e-2 | 1e-2 | 3e-3 |
| | End LR | 1e-5 | 1e-4 | 1e-4 |
| | LR warmup (epochs) | 2 | 2 | 5 |
| | Weight Decay | 1e-6 | 1e-6 | 1e-6 |
| | EMA rate | 0.999 | 0.999 | 0.999 |
| | Grad. Clipping | 5 | 2 | 150 |
| | $\log \sigma$ clipping | -10 | -10 | -10 |
| Latents | L | 8 | 8 | 28 |
| | Latent Sizes | $4 \times 14^2,$ $4 \times 7^2.$ | $4 \times 14^2,$ $4 \times 7^2.$ | $10 \times 32^2,$ $8 \times 16^2.$ $6 \times 8^2,$ $4 \times 4^2.$ |
| | Latent Width (channels) | 1 | 1 | 3 |
| | Context Size | $1 \times 7 \times 7$ | $1 \times 5 \times 5$ | $3 \times 7 \times 7$ |
| Architecture | $N_{\text{enc}}$ blocks | 3 | 3 | 4 |
| | ResBlock $C_{in}$ | 32 | 80 | 128 |
| | ResBlock $C_{hid}$ | 32 | 40 | 96 |
| | Activation | SiLU | SiLU | SiLU |
| | Likelihood | Bernoulli | Bernoulli | Discretized Logisitc |
| | # mixture comp | — | — | 10 |
| Context Prior | # Diffusion Steps | 50 | 50 | 50 |
| | # Scales in UNet | 1 | 1 | 1 |
| | # ResBlocks per Scale | 2 | 2 | 3 |
| | # Channels | 16 | 16 | 32 |
| | $\beta$ schedule | linear | linear | linear |
| | log SNR min | -6 | -6 | -10 |
| | log SNR max | 7 | 7 | 7 |

---

[4]https://github.com/openai/guided-diffusion

## D    Samples

In Figure 9, we present non-cherry-picked unconditional samples from our DVP-VAE and in Figure 10 we present non-cherry-picked unconditional samples from the pseudoinput prior $\hat{r}(\mathbf{u})$.

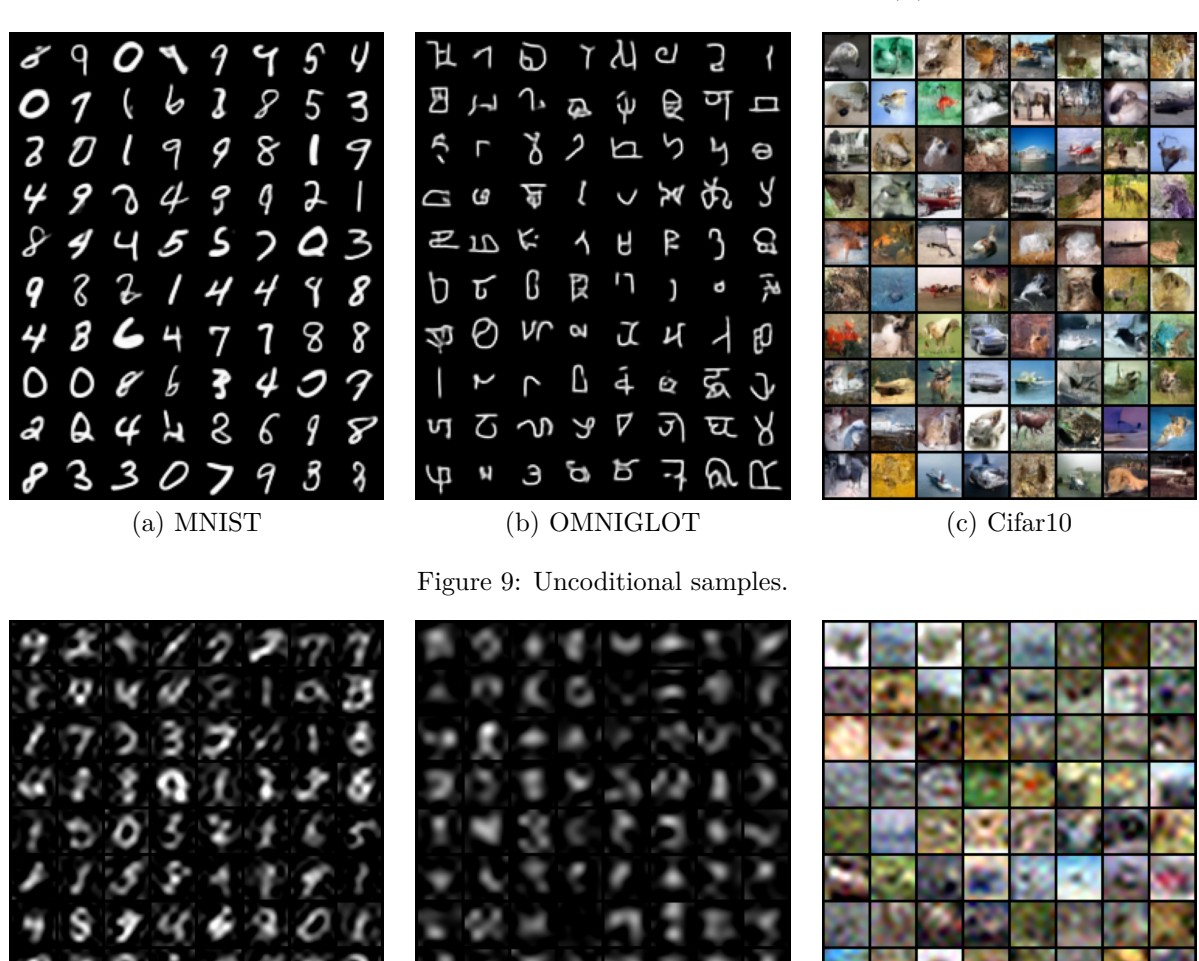

(a) MNIST                (b) OMNIGLOT                (c) Cifar10

Figure 9: Uncoditional samples.

(a) MNIST                (b) OMNIGLOT                (c) Cifar10

Figure 10: Uncoditional samples of pseudoinputs.

