# OpenReview forum: "Hierarchical VAE with a Diffusion-based VampPrior"
_TMLR — Accepted by TMLR_

### Review · Reviewer_4t7J · 2024-09-10

**Summary Of Contributions:**

The paper presents a novel approach to deep hierarchical variational autoencoders (VAEs) by amortizing VampPrior. The authors aim to enhance the performance and stability of VampPrior. They empirically validate their method on standard benchmark datasets, demonstrating improved training stability and latent space utilization.

**Audience:**

Yes

**Broader Impact Concerns:**

No broader impact concerns.

**Claims And Evidence:**

Yes

**Requested Changes:**

- Comparing wall time of training and inference
- Discussion about gain and loss, limitations. (Maybe use smaller fonts in the tables for more space.)
- Figure 2: Use only either av.pool or av.pooling for all occurrences. Use consistent font for latex and non-latex text. (Optional) use edge color instead of fill color for boxes.

**Strengths And Weaknesses:**

Strengths:
- The paper is well-structured and clearly written. The authors provide a comprehensive overview of the related work, the methodology, and the experimental results.
- The diffusion-based VampPrior provides a way of amortization that boost performance and scalability.
- The paper provides thorough empirical validation of the proposed method across multiple benchmark datasets.

Weaknesses:
- Comparing wall time of training and inference would be a plus to demonstrate the claim of improved scalability.
- Lack of substantial discussion, especially limitations.
- Figures could be polished.

---

> ### Author Response · Authors · 2024-10-19
> **Response to the review (1/2)**
>
> We thank the reviewer for the thoughtful feedback!
>
> **"Comparing wall time of training and inference would be a plus to demonstrate the claim of improved scalability."**
>
> We greatly appreciate this suggestion. The key issue with the original VampPrior formulation is its memory consumption. The authors of the VampPrior propose to use 500 or 1000 pseudoinputs for the prior, all of which should be encoded in the latent space at each training iteration (along with the batch size). This also results in a longer training time.
> That being said, if we take a VAE model size as constant, then the memory consumption for the original VampPrior will increase linearly as we increase the number of pseudoinputs (K), while for our approach it is constant. Specifically, VamPrior requires  O(DK) additional memory (with D being the size of the input and K is the number of pseudoinputs) while for DVP-VAE it is O(1) as we do not store pseudoinputs in the memory.
>
> We agree that providing a more detailed explanation and quantitative evaluation strengthens the scalability claim.  To address this, we have added extra section to the ablation study part of the paper. We have conducted experiments with the results presented in Tables 1 and 2 below. It demonstrate how the training time and memory utilization change as we increase the number of pseudoinputs/model size for VampPrior and compare these metrics  to our approach. Several key observations stand out:
> * VampPrior utilizes significantly more memory, to the extent that for the CIFAR10 dataset we were not able to fit more than 750 pseudoinputs  into a single A100 GPU. For 500 pseudoinputs, the memory utilization is already 2.5 times higher for VampPrior compared to DVP-VAE.
>
> * The wall-clock training time (seconds per epoch, the same batch size) is consistently longer for VampPrior than for our approach. Furthermore, training time increases more for VampPrior than for DVP as we increase the model size.
> * DVP-VAE exhibits slower sampling performance due to the extra step of sampling from the diffusion prior. However, we note that existing techniques (such as distillation) can significantly speed up sampling, and we leave this optimization for future work.
>
>
> Table 1a. MNIST. Experiments on a single RTX 2080 GPU, models with 32 channels width
>
> |       |    K  | Training time (s)     | GPU memory (Gb) | # Params | Sample time           |
> |-------|------|---------------|-----------------|----------|-----------------------|
> | DVP   |      |    45         |      3 Gb       | 0.7 M    |       2.2             |
> | Vamp  | 500  |    45         |      4 Gb       | 1.0 M    |       0.2             |
> |       | 1000 |    60         |      6 Gb       | 1.4 M    |       0.3             |
>
>
> Table 1b. MNIST. Experiments on a single RTX 2080 GPU, models with 64 channels width
> |       |   K   | Training time (s)   | GPU memory (Gb) | # Params | Sample time           |
> |-------|------|---------------|-----------------|----------|-----------------------|
> | DVP   |      |    50         |      4 Gb       | 2.4 M    |       2.3             |
> | Vamp  | 500  |    65         |      7 Gb       | 2.7 M    |       0.5             |
> |       | 1000 |    90         |      10 Gb      | 3.1 M    |       0.8             |
>
>
> Table 2. Cifar10. Experiments on a single A100 GPU
> |       |   K   | Training  time (s)    | GPU memory (Gb) | # Params  |Sample time           |
> |-------|------|---------------|-----------------|-----------|----------------------|
> | DVP   |      |     370       |        12 Gb    |  19.5 M   |      4.0             |
> | Vamp  | 500  |     430       |        29 Gb    |  20.5 M   |      1.5             |
> |       | 750  |     500       |        38 Gb    |  21.3 M   |      1.8             |
> |       | 1000 |    – (OOM)    |        >40 Gb   |  22.1 M   |      —            |
>
> For these experiments, we trained each model for 3 full epochs to measure the wall-clock times and memory consumption, but we did not train them to convergence.
>
> We thank the reviewer for the great experiment suggestions, we have updated the paper to include a detailed scalability discussion.

---

> > ### Author Response · Authors · 2024-10-19
> > **Response to the review (2/2)**
> >
> > **Discussion about gain and loss, limitations**
> >
> > We discuss improved models performance and better scalability in the experiments (Section 6 of the paper). We have also added the limitation section, which discuss the sampling time:
> >
> > One limitation of DVP-VAE is the sampling time. In Tables 1 and 2 above  we report sampling time (seconds per 1000 images) for DVP-VAE and VampPrior of different sizes. Long sampling time is a direct consequence of using a diffusion-based prior.  While it is a powerful generative model that allows us to achieve outstanding performance, it is known to be slow at sampling. In our experiments, we used 50 diffusion steps to generate a pseudoinput, however, there are efficient distillation techniques [1, 2] that can be applied to mitigate this issue and reduce the number of diffusion steps to just a single forward pass through the model.  We leave this optimization for future work.
> >
> > Furthermore, within DVP-VAE, we add an extra learnable block to the model, namely, the diffusion-based prior over pseudoinputs, as a result, additional modelling choices should be made. We provide an ablation study to show the effect of these choices on the performance of the model.
> >
> >
> > [1] Salimans, Tim, and Jonathan Ho. "Progressive distillation for fast sampling of diffusion models." International Conference on Learning Representations (2022).
> >
> > [2] Geng, Zhengyang, Ashwini Pokle, and J. Zico Kolter. "One-step diffusion distillation via deep equilibrium models." Advances in Neural Information Processing Systems 36 (2024).
> >
> >
> > **Figure 2**
> > Thank you for the detailed comments. We incorporated your feedback and updated Figure 2.

---

> > > ### Comment · Reviewer_4t7J · 2024-10-26
> > >
> > > Thank you for the response. My concerns have been addressed.

---

### Review · Reviewer_gcra · 2024-09-19

**Summary Of Contributions:**

This paper proposes a new hierarchical VAE, based closely on the modification of existing VAE components. In particular, the authors propose a new "Variational Mixture of Posteriors" prior (VampPrior)-like approximation of the optimal prior (i.e. the aggregated posterior), which can scale to deep hierarchical VAEs. They also propose a latent aggregation component that improves the utilization of the latent space of the VAE. These changes form the basis of the proposed hierarchical VAE; on the considered benchmark datasets of MNIST, CIFAR-10, and Omniglot, the proposed method achieves the best results of any deep hierarchical VAE.

**Audience:**

Yes

**Broader Impact Concerns:**

This work deals with the development of a hierarchical deep VAE and is fairly model-oriented. There are no ethical concerns beyond the usual ones associated with any work that deals with generative AI. I do not believe this paper requires a broader impact statement.

**Claims And Evidence:**

Yes

**Requested Changes:**

The writing in the paper is for the most part reasonable and the idea is described clearly enough; the illustrative figures are quite helpful (e.g. Figures 1 and 2). That being said, I would like the authors to tone down the "state-of-the-art" claims a bit, given the nuances I mentioned above in the "Strengths and Weaknesses" section. I would still like to provide a non-exhaustive list of minor corrections below:

- Abstract: "using much fewer parameters" -> "using fewer parameters"
- Section 2.2: "a greedy booting approach" -> "a greedy boosting approach"
- Section 4.1: Please use \citet for the Child, 2021 reference, since you are referring to the citation directly in text (as opposed to doing so parenthetically). This error occurs a few times (e.g. in Section 6.3 with "following (Hazami et al., 2022)"); please be careful with in-text citations (\citet) versus parenthetical citations (\citep or just \cite).
- Section 4.2: "At the test time" -> "At test time"
- Section 5: "using unimodal prior" -> "using a unimodal prior"
- Section 5: "Original VAE formulation" -> "The original VAE formulation"
- Section 5: "In this work, in contrary, we use" -> "In this work, in contrast, we use"
- Section 7: "with little computations overhead" -> "with little computational overhead"

**Strengths And Weaknesses:**

## Strengths

1. To the best of my knowledge, the introduced VampPrior-like class of priors is novel and the latent aggregation component really does seem to improve the utilization of the latent space of the VAE (as shown in Table 4).

2. The results of the proposed method are quite good (judging from Table 1 and Figures 3 and 4) and I am inclined to believe that this model is currently one of the best deep hierarchical VAEs in existence. However, I am not a fan of the "state-of-the-art" claims made throughout the paper. Amongst VAEs, there is at least one model that is better on CIFAR-10 (as measured by BPD), namely CR-NVAE. According to Papers with Code [a], there are also at least four diffusion models with a better BPD score.

## Weaknesses

1. Although the work has some novel elements, it ultimately relies quite a bit on existing methods. Namely, it borrows heavily from the Very Deep VAE (VDVAE) paper (Child, 2021) and the VAE with a VampPrior paper (Tomczak & Welling, 2018). In essence, one can view this paper as presenting advantageous modifications of these previous methods. This is still useful, but limits the novelty of this paper.

## Verdict

This paper presents some novel modifications to existing deep hierarchical deep VAEs. In particular, it introduces a VampPrior-like class of priors and introduces a latent aggregation component that significantly improves the utilization of the latent space of the deep VAE. The empirical results are also quite solid; amongst VAEs tested on MNIST, CIFAR-10, and Omniglot, this model is one of the best. That being said, the paper borrows heavily from Very Deep VAE (VDVAE) (Child, 2021) and VampPrior (Tomczak & Welling, 2018), which limits the novelty, somewhat. I believe the claims made in the paper are reasonably well-supported by the empirical results. I also believe this paper is of interest to at least some portion of TMLR's audience, specifically those who work on VAEs and fusions of VAEs with diffusion models. As a consequence, I recommend an accept rating for this paper.

### References

[a] https://paperswithcode.com/sota/image-generation-on-cifar-10?metric=bits%2Fdimension

---

> ### Author Response · Authors · 2024-10-19
> **Response to the review**
>
> We thank the reviewer for the thoughtful feedback!
>
> **Tone down the “state-of-the-art” claims**
>
> We acknowledge your point regarding the use of "state-of-the-art" terminology. There are several diffusion models that demonstrate better NLL values than our approach.
> We would like to highlight that the only VAE model that surpasses DVP-VAE in performance is CR-NVAE. CR-NVAE leverages the NVAE architecture and incorporates extensive data augmentation to improve its performance. In our work, we deliberately avoided using data augmentations to ensure a fair comparison with all other hierarchical VAEs. Even without augmentations, we were able to achieve better NLL than (original) NVAE and other recent hierarchical VAE models.
>
> That being said, we agree that framing our approach as "state-of-the-art" may not be fully appropriate in this context. In response to your feedback, we have revised the introduction and conclusion to temper this claim. Additionally, we have included a note in the experimental section discussing the role of data augmentations.
>
> **"Minor corrections"**
>
> Thank you for such a thorough look at our paper. We have corrected all the mentioned typos.

---

> > ### Comment · Reviewer_gcra · 2024-10-24
> > **Thank you for your response**
> >
> > Thank you for your response; I appreciate that you have made these changes!

---

### Review · Reviewer_xFeL · 2024-10-08

**Summary Of Contributions:**

The authors propose an extension of the VampPrior (Tomczak & Welling, 2018), which allows for using more latents in hierarchical VAE. Futhermore, there multiple proposed desing choices for hierarchical VAEs such as
+ Discrete cosine transform from datapoints to the mean of the pseudopoints distribution
+ Diffusion-based approximation for the pseudo-point distribution

**Audience:**

Yes

**Claims And Evidence:**

Yes

**Requested Changes:**

+ Report stds, especially in Table 1
+ Improve the writing
+ More ablation studies would be useful, e.g. break down the impact of design choices (how f modelled, prior model, etc.) in Table 1

**Strengths And Weaknesses:**

While the importance of the prior in VAEs is well-known and VampPrior has been shown to work empirically, it is still difficult to scale to deeper hierarchies, which limits the potential expressivity of the VAE. The authors address this issue. The proposed VAE method achieves good empirical performance on benchmarks while using fewer parameters than other VAE models. The ablation studies are solid, although additional studies would be beneficial.

Some weaknesses:
+ The explanation of the method is hard to follow, and could be improved. To provide a concrete example, consider the following sentence in Section 3.3: “However, unlike iVAE, we do not treat u as an observed variable. Instead, we introduce the pseudo-input prior distribution pγ(u) with learnable parameters γ. “
This is confusing for the reader. Based on the equation (9), it looks like u is a noisy version of x, where mean is given by deterministic mapping f. There are no trainable hyperparameters in f, which is also hinted by the sentence “the first important component in our approach is the form of the non-trainable transformation from the input to the pseudoinput space.”, so for the reader is unclear what is γ? Further, in the equation (15), there is no dependence on γ as hinted by the ELBO arguments. The authors later explain γ, and how the pseudo-point distribution is approximated, but the damage is already done: the story is hard to follow.
+ There are many separate design proposal on top of hierarchical VAEs. To understand impact of these c, ablation studies are very useful. (thanks of having some in the manuscript). It would be useful to understand which of these are the most important in explaining the empirical performance in Table 1.
+ Please report a measure of variation in the experimental results (Table 1). This is crucial as there are quite few trials (experiments replicated over 4 seeds)

---

> ### Author Response · Authors · 2024-10-19
> **Response to the review**
>
> We thank the reviewer for thoughtful feedback!
>
> **"Report stds, especially in Table 1."**
>
> We would like to highlight that Table 5 and Figure 5 already contain standard deviations for those experiments. For MNIST, the standard deviation is very low, making it hardly visible in Figure 5 (a). We have omitted std in Table 1 for compactness, and to not repeat it. But we agree that it would be better to include those in both cases. We, therefore, updated Table 1 to contain standard deviations as well.
>
> Unfortunately, it is not common in the literature to report the test performance of the generative models on several random seeds. As Table 1 shows it, none of the compared approaches reported standard deviation. This mainly happens due to the high computational complexity of each experiment. Nevertheless, we decided to run multiple trials at least on the smaller datasets like MNIST and OMNIGLOT.
>
> **"Improve the writing"**
>
> Thank you for this feedback. To clarify, in our approach, the variable $u$ is not treated as an observable during inference; rather, we suggest sampling it from the prior. However, during training, $u$  is indeed obtained as a non-trainable noisy transformation of the input $x$. We have revised Section 3.3 to make this point clearer and more easily understood.
>
> **"More ablation studies would be useful, e.g. break down the impact of design choices (how f modelled, prior model, etc.) in Table 1"**
>
> We have conducted the following ablations studies of the results presented in Table 1:
> * The effect of diffusion-base prior (Table 5)
> * The effect of different f: we compare DCT and Downsampling (Figure 5)
> * The effect of vamp-prior pseudo-input  priors: Diffusion and Mixture of Gaussians (Figure 5)
> * The effect of different pseudoinput sizes: 3 to 11 (Figure 5)
>
> However, some design choices are interdependent and must be used together: for instance, f function and the prior type (we cannot use one without another). Thus, an exhaustive ablation study is not really possible. Nevertheless, we are open to any specific suggestions you may have for further potential improvements and would be happy to explore them.

---

> > ### Comment · Reviewer_xFeL · 2024-11-04
> >
> > Thank you for the response. Regarding the ablation studies, fair enough—some design choices are indeed interdependent. I have no further concerns.

---

### Author Response · Authors · 2024-10-19
**General response**

We would like to thank all reviewers for their time and effort in reading and evaluating our paper. We have put a lot of effort into addressing all of your questions in individual comments. We have also updated the manuscript and highlighted all the changes in orange.

The following changes were made to the manuscript:
* We added an ablation study comparing training time and memory consumption of DVP-VAE with the original VampPrior.
* We added the “Limitations” section.
* We made several minor corrections (i.e., updated Figure 2, improved writing is section 3, and toned down “state-of-the-art” claims).

---

### Decision · Action_Editor_JVQh · 2024-11-07

**Recommendation:** Accept as is

**Comment:**

The paper is in general solid piece of research, and the reviewers had clear consensus on both its strengths and weaknesses. The technical solution is new and sound and it was found to work well in practice. However, the work is somewhat incremental and building heavily on previous deep VAE papers, the writing could still be improved, and the impact is reduced by limited discussion. Nevertheless, the paper is clearly suitable for publication in TMLR.

**Audience:**

Variational autoencoders are an active research topic in the field and the work is likely to be interesting for a sub-community of substantial side, both from a methodological and empirical perspective.

**Claims And Evidence:**

The paper introduces specific extensions for VampPrior. The new contributions are explicitly laid out, and the main claim regarding performance is validated with clear empirical experiments.